# InterAvatar: Real-time Interactive Portrait Animation via Behavioral Interaction Prompts

## Abstract

Portrait animation can generate realistic animated videos from static portrait images and play a crucial role in a wide range of real-world applications. Despite substantial advances in realism, existing portrait animation methods suffer from critical limitations: slow inference speeds unsuitable for interactive scenarios and the absence of behavioral interaction capabilities, significantly restricting immersive user experiences. To address these limitations, we propose InterAvatar, the first framework that adapts a real-time video diffusion transformer for portrait animation conditioned on behavioral interaction prompts. Specifically, InterAvatar is built upon the diffusion transformer Wan2.1-1.3B and LTX-Video-2B with diffusion distillation. It is conditioned on a reference image, audio signals, and behavioral interaction prompts to animate avatars. To enhance appearance consistency and reduce drifting in real-time animation frameworks, we introduce a representation decoupling strategy that separates identity and attribute information from the reference appearance. We also present the first work to introduce behavioral interaction prompts into portrait animation, proposing pioneering strategies for encoding and injecting these prompts into diffusion transformers. Besides, we introduce a hybrid data curation pipeline for systematically collecting, annotating, and filtering real and synthetic video data annotated with behavioral interaction prompts. Extensive evaluations on HDTF, CelebV-HQ, and RAVDESS demonstrate InterAvatar achieves comparable video quality with state-of-the-art models and effectively simulates realistic behavioral interactions, enhancing the interactive user experience. InterAvatar can generate 80 video frames at 512×512 resolution in just 5 seconds on an Nvidia H800 GPU, offering an optimal balance between accuracy and efficiency.

## 1 Introduction

Portrait animation refers to the task of generating a realistic animated video of a human face or character from a static portrait image. This technology is significant for a broad range of applications, including entertainment, augmented reality, virtual assistants, and social media. Existing portrait animation methods have made remarkable progress in realism and fidelity. Audio-driven approaches animate a still face to match a speech soundtrack, producing high-resolution talking-head videos with accurate lip-sync and expressive facial movements. Other methods drive portraits using explicit motion signals. More recently, video diffusion transformers have been applied to portrait animation (Cui et al., 2024; Wang et al., 2025b; Lin et al., 2025), leveraging their powerful generative capabilities to produce avatars with dynamic backgrounds and high visual fidelity.

Despite these advances, current DiT-based approaches exhibit significant limitations, lacking necessary interactivity between the portrait animation process and the user: (1) Most state-of-the-art methods suffer from slow inference speeds, forcing users to endure considerable waiting times before viewing the generated animations. However, users generally expect immediate visual feedback upon interaction. Such delays disrupt the perception of direct manipulation and negatively impact user experience. For example, Wan2.1-14B (Wang et al., 2025a) requires roughly 30 minutes to synthesize a 5-second clip on an H800 GPU, which results in unacceptable latency for interactive scenarios. (2) Current portrait animation frameworks only allow plain interaction (e.g., text prompt

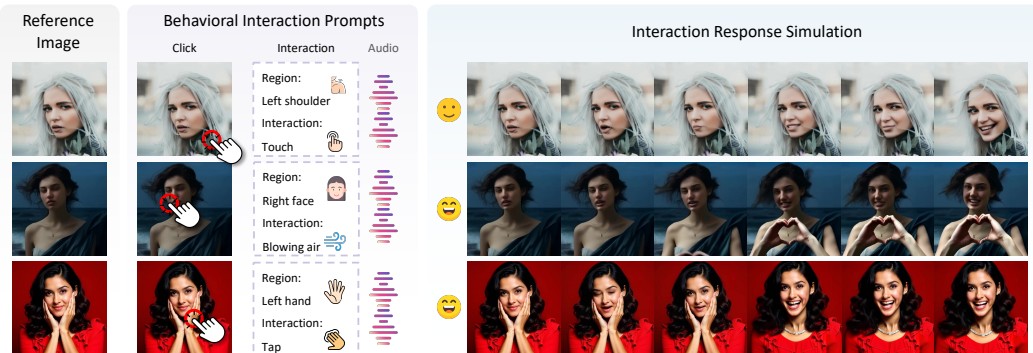

Figure 1: **The generation results of InterAvatar with different behavioral interaction prompts**. When InterAvatar receives customized behavioral interaction prompts from the user, it can simulate natural human responses based on the context and animate the avatar in real time.

and audio signals) between users and avatars but inherently lack the behavioral interaction capabilities present in real-world scenarios. Behavioral interaction between users and avatars is crucial because it significantly enhances user engagement, immersion, and perceived realism by enabling intuitive tactile interactions, similar to those encountered in daily life. For example, gently touching an animated avatar's cheek could trigger realistic facial responses, such as subtle smiling, blushing, or eye movements, which deepens the emotional connection between users and avatars. Therefore, simulating behavioral interactions within portrait animation represents an important direction for advancing interactive realism and improving user interaction experience. However, the paradigm of incorporating such interactions into portrait animation remains unexplored.

To address these limitations and enrich user interactivity, we propose the **Inter**active **Avatar** synthesis framework (InterAvatar). InterAvatar adapts a pretrained video diffusion transformer (HaCohen et al., 2024) for multi-condition generation, receiving a reference image, audio signals, and behavioral interaction prompts as inputs to animate portraits in real-time. We are the first to investigate how to adapt a pretrained video diffusion transformer for real-time portrait animation tasks conditioned on behavioral interaction prompt inputs. We observe that, although Wan2.1-1.3B and LTX-Video-2B with diffusion distillation (Huang et al., 2025) support real-time generation, they both suffer from poor appearance consistency and drifting compared to larger models. By decoupling the appearance representation of the reference image into distinct identity and attribute components, and injecting corresponding knowledge from their respective vision encoders into the denoising process, our approach substantially enhances identity preservation and reduces drifting artifacts. In addition, we introduce the first attempt to simulate behavioral interactions between users and avatars within portrait animation. Specifically, we interpret user clicks as a form of interaction position inputs: after specifying a behavioral interaction type, the user clicks on the avatar's body or face region in the reference image to simulate a behavioral interaction. We regard these user interaction positions and interaction types as behavioral interaction prompts, and investigate effective strategies to encode and incorporate these prompts for realistic interaction simulation. Lastly, we introduce a hybrid annotation pipeline that combines real and synthetic videos with behavioral interaction prompts, enabling the model to learn this special single-avatar–to–invisible-user interaction manner while ensuring the generation of contextually appropriate human reactions. Evaluations on mainstream benchmarks, including HDTF (Zhang et al., 2021), CelebV-HQ (Zhu et al., 2022), and RAVDESS (Livingstone & Russo, 2019), demonstrate that InterAvatar achieves comparable video quality to state-of-the-art models, with real-time generation capabilities. Specifically, InterAvatar can generate a 5-second video at 512×512 resolution in just 5 seconds on an Nvidia H800 GPU, offering an optimal balance between accuracy and efficiency. Through visualizations, we also illustrate that our interactive portrait animation effectively simulates behavioral interactions, improving user interactivity.

The **contributions** of this work are three-fold: **(1)** We explore how to adapt a pretrained video diffusion transformer for real-time portrait animation and propose a novel method that decouples identity and attribute representations to enhance appearance consistency. **(2)** We present the first work to introduce behavioral interaction prompts into portrait animation, proposing effective strategies for encoding and injecting these interaction prompts into diffusion transformers. **(3)** We propose a data curation pipeline designed to systematically collect, filter, and annotate real videos from ex-

isting datasets, as well as generate and filter synthetic videos annotated with behavioral interaction prompts to effectively support interactive portrait animation training.

## 2 RELATED WORK

### 2.1 PORTRAIT ANIMATION

Early works on human image animation leveraged warping techniques and GAN-based frameworks (Goodfellow et al., 2020). Some methods perform unsupervised keypoint-based motion transfer (Siarohin et al., 2019; Chan et al., 2019), others utilize pose-conditioned generation (Wang et al., 2019), while motion-appearance disentanglement further improves fidelity (Siarohin et al., 2021). With the recent advances of diffusion probabilistic models (Ho et al., 2020; Nichol & Dhariwal, 2021), human animation has experienced a substantial leap in realism, temporal stability, and controllability. Several works have adopted diffusion models for both pose-driven human animation, where human motion is represented as 2D skeletons or 3D meshes, and regional animation, which enables local control. Recently, audio-driven human animation (Tian et al., 2024; Ji et al., 2024; Meng et al., 2024) has gained increasing attention, where speech signals control facial and body motion. Hallo adopts an end-to-end diffusion paradigm for audio-driven portrait animation, with the latest Hallo3 (Cui et al., 2024) introducing a video diffusion transformer to better handle dynamic motions and complex backgrounds. Pushing generalization further, OmniHuman-1 (Lin et al., 2025) proposes a scalable multimodal framework trained on mixed conditioning signals, enabling high-fidelity animation. Different from previous methods, we propose a real-time human animation framework enabling physical interaction prompts.

### 2.2 INTERACTIVE VIDEO GENERATION

Interactive video generation combines high-quality video synthesis with user-driven control and responsive feedback mechanisms. Recently, a key trend is the integration of strong video generative models into closed-loop frameworks that respond to control signals in real time. Numerous works treat the video model as a learnable simulator that can be steered by user actions or policies. For example, diffusion-driven "world models" have been used as game engines and interactive environment simulators (Valevski et al., 2024; Che et al., 2024), enabling an agent or player to influence the generated video on the fly. Another line of work focuses on controllable video generation for human-centric content. Researchers have explored methods to explicitly guide the motion and appearance in generated videos using high-level conditions. For instance, Follow-Your-Click (Ma et al., 2024) enables precise and intuitive local animation by allowing users to animate specific regions with a simple click and a short motion prompt. Besides, playable video techniques learn a latent video representation that users can manipulate, enabling interactive animation of objects or human characters in a synthesized scene (Menapace et al., 2021; 2022).

## 3 METHOD

### 3.1 OVERVIEW

The InterAvatar framework consists of two core components: (1) a video diffusion transformer backbone and (2) a multi-condition encoding and injection mechanism that enables controllable interactive human animation. Our architecture leverages the pretrained Wan2.1-1.3B and LTX-Video-2B as the baselines. To achieve interactive compatibility and improved controllability, our approach integrates multiple conditioning modalities, including reference images, audio signals, and behavioral interaction prompts directly within the diffusion-based denoising pipeline. Notably, our method introduces a novel capability in human animation: simulating realistic behavioral interactions between users and the avatar guided explicitly by behavioral interaction prompts. We structure our discussion as follows: In Section 3.2, we revisit the architecture of the video diffusion transformer backbone. In Section 3.3, we detail the encoding strategies employed for several conditioning signals. Finally, Section 3.4 details the encoding and injection strategies for behavioral interaction prompts.

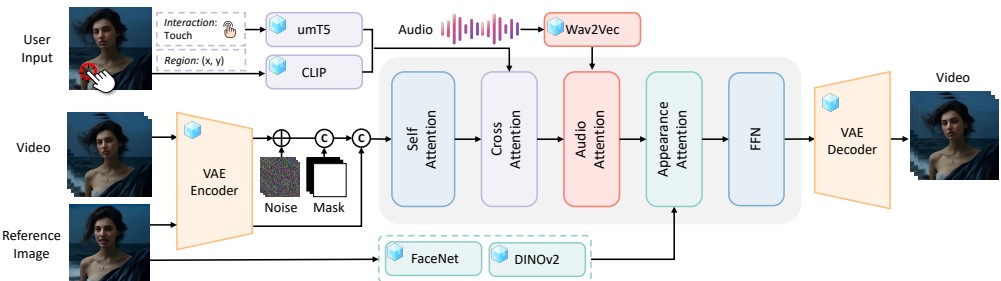

Figure 2: Architectures of InterAvatar with Wan2.1. We omit the time embedding for simplicity.

## 3.2 VIDEO DIFFUSION TRANSFORMER

**Inputs.** Initially, video frames are compressed using the VAE, which aggressively downsamples spatial and temporal dimensions to yield a compact sequence of latent patch tokens. At the beginning of the diffusion process, Gaussian noise is added to these latent tokens, and the embedded diffusion timestep $t$ is supplied to the transformer model. Additionally, the denoising transformer receives multimodal conditioning signals: (1) a text prompt, encoded by a pretrained umT5 (Chung et al., 2023) text encoder into text embedding tokens that provide semantic guidance via cross-attention; and (2) an image conditioning tensor $\mathbf{z}_{\text{img}}$, obtained by encoding the first frame through the VAE, which is injected into the latent sequence to enable image-to-video generation; and (3) a binary mask $M$, indicating preserved versus to-be-generated frames, which explicitly specifies the temporal regions conditioned or synthesized. Furthermore, CLIP (Radford et al., 2021) features extracted from the first frame serve as a global context, fused into the model via decoupled cross-attention.

**Transformer Design.** The transformer backbone extends the original DiT architecture (Peebles & Xie, 2023) by incorporating a spatio-temporal attention and decoupled cross-attention layers in each transformer block. Latent video representations, together with rearranged mask tokens, are flattened into sequential tokens processed by multi-head self-attention layers to capture spatial and temporal dependencies. To encode positional information within the attention mechanism, we employ 3D Rotary Position Embeddings (RoPE) (Su et al., 2024), which explicitly capture each token's spatial coordinates and temporal frame indices. Transformer layers alternate between self-attention on the video tokens and cross-attention to the provided context embeddings. Following the attention layers, feed-forward networks (FFN) are included to enhance representations, and RMS normalization (Zhang & Sennrich, 2019) is employed to enhance training stability. After performing $T$ diffusion steps, the transformer outputs denoised latent tokens that are decoded by the VAE decoder.

## 3.3 CONTROLLING CONDITIONS

**Reference Image Conditioning.** Early approaches (Tian et al., 2024; Cui et al., 2024) typically used a trainable copy of the backbone network as an additional reference network to encode the appearance representations of the reference image. However, this paradigm substantially increases both the model complexity and training overhead. Leveraging the inherent flexibility of the transformer architecture, we observe that directly appending latent representations of the reference image before the noisy latents effectively preserves reference appearance coherence across extended video sequences. Similar observations have also been confirmed in large-scale models trained on extensive datasets (Lin et al., 2025). Additionally, we observe the small baseline model exhibits poor appearance consistency and temporal drifting compared to larger models over extended durations. To address such limitation, we extract and decompose human-centric representations from the reference images into separate identity and attribute embeddings. The identity embedding is generated by cropping facial regions and encoding them using the InsightFace (Ren et al., 2023) model, which can ensure robust extraction of facial identity features. To capture detailed facial expressions and gestures representations, the attribute embedding is derived using the self-supervised vision encoder DINOv2 (Oquab et al., 2023), which has demonstrated effectiveness in preserving nuanced subject-specific details. Both embeddings are integrated into our diffusion transformer through an appearance attention module. Given the input latents $\mathbf{z}_{\text{in}}$, identity embeddings $\mathbf{c}_{\text{id}}$, and attribute

embeddings $\mathbf{c}_{\text{att}}$, the output features $\mathbf{z}_{\text{out}}$ are computed as follows:

$$\mathbf{z}_{\text{out}} = \mathbf{z}_{\text{in}} + \text{Proj}\left(\text{Softmax}\left(\frac{\mathbf{Q}\mathbf{K}_{\text{id}}^T}{\sqrt{d}}\right)\mathbf{V}_{\text{id}} + \text{Softmax}\left(\frac{\mathbf{Q}\mathbf{K}_{\text{att}}^T}{\sqrt{d}}\right)\mathbf{V}_{\text{att}}\right), \tag{1}$$

where $\mathbf{Q} = \mathbf{z}_{\text{in}}\mathbf{W}_q$, $\mathbf{K} = \mathbf{c}\mathbf{W}_k$, and $\mathbf{V} = \mathbf{c}\mathbf{W}_v$, we omit the subscripts *id* and *att* for simplicity. The term Proj denotes a linear projection layer whose parameters are initialized to zero to stabilize training. This decoupled appearance conditioning enhances the model's ability to preserve facial identity, ensure high fidelity, and achieve appearance coherence throughout extended video sequences.

**Audio Conditioning.** To enable precise control over lip movements and facilitate highly realistic portrait animation, we incorporate audio conditioning. In practice, we leverage a pretrained wav2vec (Baevski et al., 2020) network to encode the input audio waveform into a sequence of latent audio features. The extracted audio features are initially aligned with video frames by linear interpolation such that each video frame corresponds to a single audio embedding. To align the audio embeddings with the temporal resolution of the compressed video latents, we downsample the audio features using a convolutional network comprising two layers, each with a kernel size of 2. These downsampled audio features are subsequently attended by the video tokens through audio cross-attention. To be specific, both the video latents and audio features are reshaped into frame-wise representations, enabling the model to perform frame-wise cross-attention.

### 3.4 BEHAVIORAL INTERACTION PROMPT

To enhance user interactivity and realistically simulate interactions between users and generated avatars, we introduce behavioral interaction prompts. These behavioral interaction prompts specify desired interactions with particular body regions of the animated human, enabling a video diffusion model to generate content that accurately reflects natural behavioral responses. Each behavioral interaction prompt comprises two key components: (1) spatial coordinates $(x, y)$ marking a precise location on the targeted region within a reference image, (2) a linguistic interaction description that briefly describe the action type of interaction.

**Interaction Position Processing.** We explore four approaches for encoding the spatial information of the behavioral interaction prompt: (1) positional encoding (Ravi et al., 2024) of spatial coordinates, (2) binary regional masks created by defining a local region centered around the interaction point, (3) cropping a local regional view around the interaction point, and (4) drawing a small, colored marker at interaction coordinate. Realistic simulation of behavioral interactions requires not only knowing the interaction's region and type but also understanding the broader context, including the human's current activity and any objects held. While all methods can highlight the positional information of the interaction region, the last approach additionally preserves contextual information from the rest of the image. Therefore, we adopt a hybrid strategy that combines (1) positional encoding and (4) marker-based annotation, allowing the model to encode precise interaction coordinates while retaining the necessary image context. First, we obtain the positional encoding of the normalized coordinates $(x, y)$, which is projected through a multi-layer perceptron (MLP) to form positional tokens. Next, leveraging the visual prompting capability of the CLIP encoder (Shtedritski et al., 2023), we overlay a small red marker at the interaction region in the input image. This guides the CLIP vision encoder's attention specifically toward the interaction region while preserving high-level image context comprehension. Finally, the positional tokens are injected with CLIP visual features through cross-attention layers, enabling joint injection of behavioral interaction prompts.

**Interaction Description Processing.** To more precisely condition the diffusion model on specific interaction types, we incorporate concise textual descriptions of behavioral prompts. Each interaction is represented by a simple keyword such as touch, poke, or similar action terms, which capture the intended interaction style. Different interaction types naturally lead to distinct avatar responses, for instance, aggressive actions provoke intense reactions, whereas friendly actions elicit more amiable ones. Similar to us, prior work (Ma et al., 2024) conditions the model with additional text prompts, but encodes them through dedicated adapter layers, whereas we directly leverage the diffusion transformer's inherent text comprehension capability. Specifically, the interaction keyword is expanded into a natural language sentence following the template "*The person is reacting right after being* `<interaction>`." This sentence is prepended to the user prompt (separated by a line break), and the combined prompt is then encoded by the pretrained text encoder. The resulting embeddings are injected into the diffusion transformer through original cross-attention layers.

# 4 DATASET CURATION

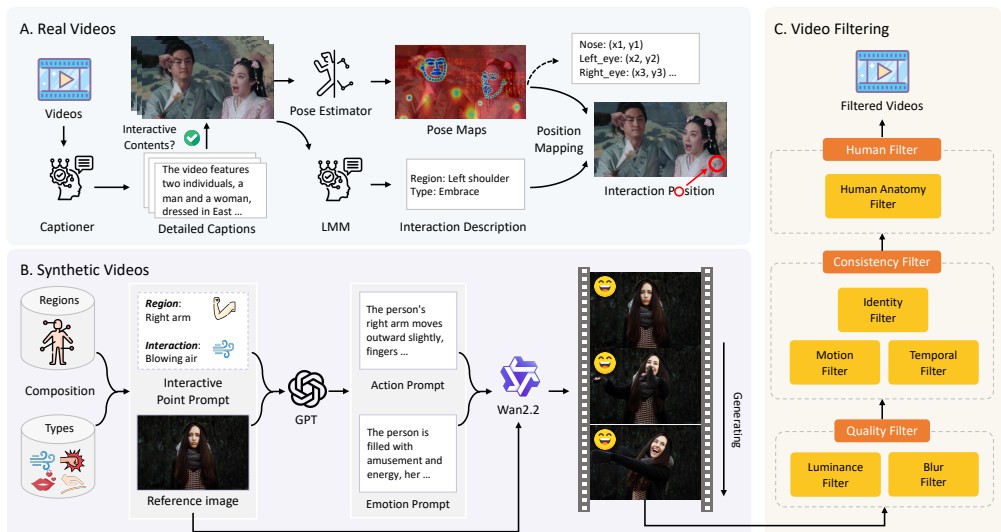

Figure 3: Data curation pipeline for videos with behavioral interaction prompts.

## 4.1 DATA SOURCE

Our training dataset comprises two main components: audio-driven portrait animation videos and videos with behavioral interaction prompts. For the first part, we aggregate several publicly available sources, including Hallo3 (Cui et al., 2024), Casual Conversations v2 (CCv2) (Porgali et al., 2023), CelebV-Text (Yu et al., 2023), and MEAD (Wang et al., 2020). Hallo3 is a recent dataset of high-quality talking-head videos with English speech, capturing diverse head poses, expressions, and backgrounds. CCv2 provides 26K video monologues from 5.5K participants recorded in a half-body view, which we include to increase the diversity of portrait and speaking styles in our training data. MEAD dataset is a high-quality emotional audio-visual corpus featuring 60 actors expressing eight emotions. To collect real videos with behavioral interaction prompts, we rely on the Open-HumanVid (Li et al., 2024) dataset, a large-scale human-centric video collection. OpenHumanVid consists of millions of high-quality video clips of people performing various activities, complete with descriptive captions. Specifically, we filtered and selected approximately 30K real videos with behavioral interaction prompts from the OpenHumanVid dataset. In addition, we augment our training set with 50K synthetic interaction videos generated by our elaborate data curation pipeline.

## 4.2 DATA ANNOTATION

Our proposed behavioral interaction prompts require a generated *single* avatar to react to an *invisible* user positioned behind the camera, based on a specified interaction type and region. This setup departs significantly from conventional interaction contents, where *multiple visible* individuals interact in the same scene. Therefore, publicly available videos rarely capture scenarios in which a person interacts with an unseen partner behind the camera. To overcome this data scarcity, we introduce a hybrid annotation pipeline that integrates both real and synthetic videos as training data. This design enables our framework to learn the unique interaction pattern from synthetic data while maintaining capabilities to generate high-fidelity and contextually appropriate human reactions through real data.

**Annotation for Real Videos.** To construct behavioral interaction prompts from OpenHumanVid, we design an automated annotation pipeline. First, we use Qwen2.5-VL-7B (Bai et al., 2025) to generate descriptive captions for each video, focusing on human-centric content. Videos are then filtered to retain only those involving at most two people, covering both single- and dual-person interactions. Next, Qwen2.5-VL-7B is applied again to infer the interaction type and the corresponding body region. For example, when a hand touches a head, the labels are "touch" (type) and "head" (region). However, the large multimodal model (LMM) often lacks fine-grained accuracy in localizing body regions. To mitigate this, we employ the pose estimator DWPose (Yang et al., 2023) on the first

Table 1: Quantitative comparisons with audio-conditioned portrait animation baselines. The best score is in  blue , with the second-best score in  green .

| Methods | HDTF | | | | | CelebV-HQ | | | | | RAVDESS | | | | |
|---|---|---|---|---|---|---|---|---|---|---|---|---|---|---|---|
| | IDC↑ | ASE↑ | Sync-C↑ | FID↓ | FVD↓ | IDC↑ | ASE↑ | Sync-C↑ | FID↓ | FVD↓ | IDC↑ | ASE↑ | Sync-C↑ | FID↓ | FVD↓ |
| SadTalker Zhang et al. (2023b) | 0.845 | 5.500 | 1.645 | 88.026 | 129.474 | 0.851 | 5.383 | 2.123 | 69.411 | 167.184 | 0.859 | 5.134 | 3.553 | 91.426 | 217.972 |
| Hallo Xu et al. (2024) | 0.980 | 5.387 | 1.874 | 14.108 | 138.088 | 0.931 | 5.072 | 2.253 | 31.922 | 178.818 | 0.940 | 4.231 | 2.703 | 28.852 | 288.555 |
| AniPortrait Wei et al. (2024) | 0.977 | 5.442 | 1.324 | 16.937 | 146.850 | 0.935 | 5.190 | 1.218 | 33.229 | 228.409 | 0.920 | 4.273 | 1.969 | 29.946 | 614.112 |
| DreamTalk Zhang et al. (2023a) | 0.823 | 5.252 | 1.558 | 111.251 | 182.523 | 0.776 | 4.806 | 1.925 | 99.309 | 210.877 | 0.838 | 5.058 | 3.470 | 117.123 | 343.260 |
| V-Express Wang et al. (2024) | 0.971 | 5.176 | 1.972 | 23.228 | 114.666 | 0.848 | 4.829 | 1.896 | 55.500 | 161.924 | 0.936 | 4.165 | 3.548 | 29.123 | 222.759 |
| EchoMimic Chen et al. (2025) | 0.914 | 5.345 | 1.659 | 56.151 | 176.281 | 0.878 | 5.102 | 1.821 | 52.645 | 192.203 | 0.879 | 4.820 | 3.430 | 75.383 | 268.632 |
| Hallo3 Cui et al. (2024) | 0.975 | 5.172 | 1.935 | 15.820 | 115.060 | 0.927 | 4.822 | 2.297 | 29.496 | 168.635 | 0.947 | 4.869 | 3.860 | 23.091 | 237.704 |
| **InterAvatar-LTX** | 0.981 | 5.390 | 1.655 | 13.725 | 107.719 | 0.940 | 5.164 | 2.035 | 32.636 | 162.301 | 0.975 | 4.888 | 3.536 | 12.394 | 191.570 |
| **InterAvatar** | 0.981 | 5.392 | 6.072 | 11.171 | 87.680 | 0.932 | 5.203 | 3.283 | 28.891 | 176.115 | 0.970 | 4.835 | 4.637 | 11.243 | 76.529 |

frame to extract keypoint coordinates and map them to the predicted region label. In multi-person scenarios, the LMM is further used to identify which individual corresponds to the interaction region. Finally, the localized region coordinates at the first frame and the region label are combined to form the annotated behavioral interaction prompt.

**Annotation for Synthetic Videos.** To supplement the training corpus, we construct additional data with behavioral interaction prompts using a synthetic generation pipeline. We begin by collecting 8K high-quality human portrait images, all compliant with copyright regulations. For each image, we apply DWPose and Qwen2.5-VL-7B to identify visible body regions that can serve as interaction targets, along with their corresponding coordinates. The image, selected body regions, and predefined interaction types are then provided to the GPT (Hurst et al., 2024) model, which produces contextually relevant responses consisting of action prompts and emotion prompts (see Appendix B.1 for details). Given the reference image together with the action and emotion prompts, we utilize the Wan2.2-I2V-A14B model (Wang et al., 2025a) to synthesize a video clip, where the animated avatar enacts the specified physical action and emotion, with the reference image serving as the initial frame. This pipeline generates high-quality synthetic videos for diverse interaction scenarios, which are integrated into the training set to help the model adapt to this unique interaction pattern.

## 4.3 Data Filtering

All collected and generated videos are passed through a comprehensive processing pipeline to ensure both quality and consistency. For videos containing audio, we follow the preprocessing procedures established in the Hallo3 pipeline (see Appendix B.2 for details). In addition, we incorporate an LMM-based automatic filtering stage to detect and exclude undesirable artifacts such as watermarks, subtitles, multiple individuals in the frame, facial occlusions, and other visual distortions. Synthetic videos may exhibit distinctive artifacts introduced by generation models, posing further challenges that can compromise training quality. To mitigate this, we apply several specialized filtering procedures. First, a video quality filter evaluates metrics such as luminance and blur to discard low-quality clips. Second, we compute motion, temporal consistency, and identiy consistency scores to remove invalid or unstable videos. Motion scores are obtained using CoTracker (Karaev et al., 2024), while temporal consistency and identity consistency are computed as the average CLIP feature and InsightFace (Ren et al., 2023) feature similarity scores between adjacent frames. Finally, a human quality filter assesses anatomical correctness and filters out content with distorted limbs. Together, these procedures yield a clean and reliable set of videos for training.

## 5 Experiments

### 5.1 Main Results

Implementation details are provided in Appendix C, while additional qualitative results, ablation studies, and generated videos are available in Appendix D and the Supplementary Materials.

**Comparisons with Existing Methods.** We compare InterAvatar with other state-of-the-art methods in Table 1. Following previous works, we randomly select 100 test samples for each dataset. For quick evaluation, we use the same prompt, "the person is talking," for each sample. However, this simplification may impair InterAvatar's performance, as the training captions generated by LMMs are typically more detailed. Overall, InterAvatar achieves the best performance in terms of IDC, FID,

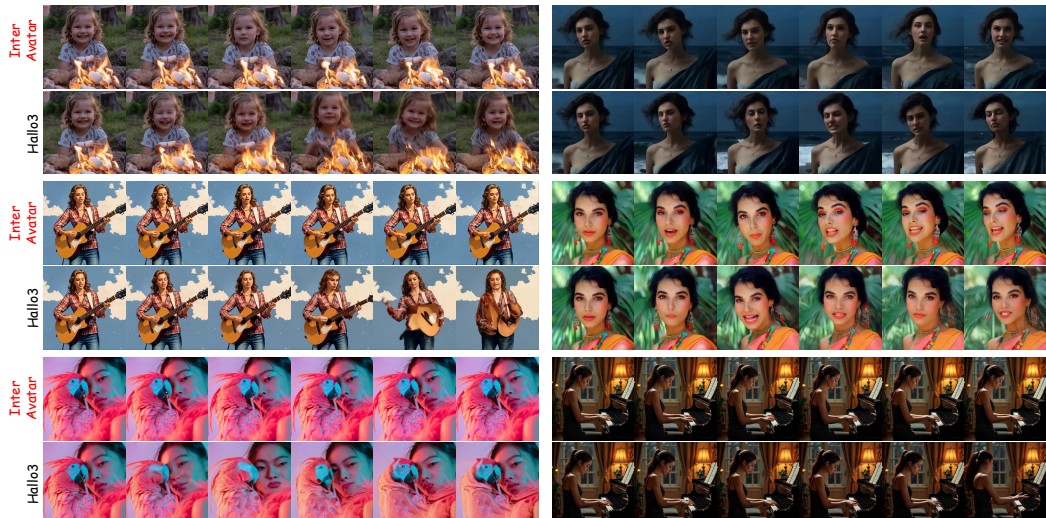

Figure 4: Comparison on scenes with dynamic foreground, background, and interactive subjects.

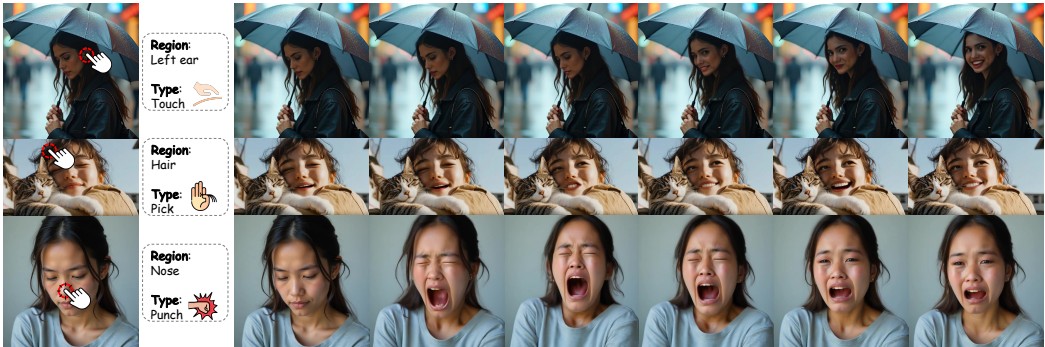

Figure 5: Qualitative interactive human animation results.

and FVD, highlighting its superior visual fidelity and temporal consistency. Additionally, our default version with Wan2.1-1.3B achieves state-of-the-art lip-sync quality while preserving real-time generation capability. We also provide qualitative comparisons in Appendix D.

**Comparisons with video DiT-based Methods.** Video DiT-based animation methods excel at generating dynamic foregrounds and backgrounds in animated videos. We compare our approach with another video DiT-based method, Hallo3, as illustrated in Figure 4. Our method demonstrates superior preservation of appearance consistency, including characters, backgrounds, and interactive objects, as well as smoother and more natural motion dynamics. Notably, our inference speed significantly outperforms Hallo3, achieving more than $30\times$ faster inference at the same resolution.

**Qualitative Interaction Visualizations.** We provide qualitative evaluations in Figure 5. InterAvatar can animate avatars to simulate interactive responses conditioned on various input behavioral interaction prompts, including specified interaction regions and interaction types. For example, in the third row, when the user selects the woman's nose with a "punch" interaction prompt, her expression shifts to one of pain, followed by a crying reaction.

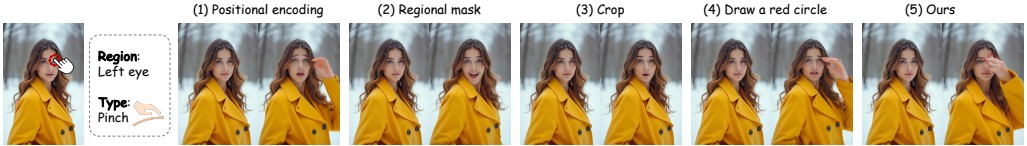

Figure 6: Comparisons of different interaction position encoding strategies.

Table 2: Conditioning methods

| Method | IDC↑ | Sync-C↑ | FID↓ | FVD↓ |
|---|---|---|---|---|
| Add | 0.970 | 5.958 | 13.942 | 115.184 |
| Refnet | 0.979 | 6.143 | 13.755 | 101.421 |
| **Concat** | 0.981 | 6.072 | 11.171 | 87.680 |

Table 3: Appearance encoding

| Method | IDC↑ | Sync-C↑ | FID↓ | FVD↓ |
|---|---|---|---|---|
| Identity | 0.972 | 6.036 | 14.250 | 121.114 |
| Attribute | 0.967 | 6.085 | 12.513 | 94.065 |
| **Both** | 0.981 | 6.072 | 11.171 | 87.680 |

Table 4: Appearance injection

| Method | IDC↑ | Sync-C↑ | FID↓ | FVD↓ |
|---|---|---|---|---|
| adaLN-zero | 0.965 | 5.395 | 15.694 | 137.325 |
| Separate | 0.983 | 6.044 | 11.205 | 90.523 |
| **Decoupled** | 0.981 | 6.072 | 11.171 | 87.680 |

Figure 7: Comparisons of different data annotation strategies.

## 5.2 ABLATION STUDY

**Reference Image Conditioning.** As shown in Table 2, concatenating the reference image latents with the noisy latents achieves performance comparable to using a reference network. However, the reference network significantly increases the parameter count of the backbone and incurs additional computational overhead. We also evaluate an alternative approach that adds the reference image latents to the latent of each frame, which results in lower performance than concatenation. Therefore, we adopt the simpler strategy to maintain a balance between efficiency and effectiveness.

**Appearance Encoding.** We evaluate three methods for appearance encoding, as shown in Table 3. Using only the identity embedding or only the attribute embedding results in performance degradation. In contrast, our proposed decoupling strategy, which integrates both identity and attribute embeddings, achieves the best performance in terms of appearance fidelity and expressiveness.

**Appearance Injection.** We compare three methods for injecting appearance features in Table 4. adaLN-zero (Peebles & Xie, 2023) exhibits limited modulation capacity, while employing two separate cross-attention modules increases model complexity without notable improvements. The adopted decoupled cross-attention mechanism provides a more effective and efficient way to inject identity and attribute information separately, bringing superior visual appearance quality.

**Interaction Prompt Position Encoding.** We consider four strategies for encoding the position of behavioral interaction prompts in Section 3.4. We visualize an example where a person's left eye is pinched, and compare the generated responses across encoding methods in Figure 6. Our adopted strategy generates a reaction in which the person touches the interaction region with her hand, resulting in a more natural and contextually appropriate response compared to other methods.

**Hybrid Data Annotation.** We ablate our hybrid data annotation method in Figure 7. Training solely on real data ensures high visual fidelity but lacks interactivity. Conversely, relying only on synthetic data enables the model to learn richer user–avatar interaction patterns, yet the generated avatars may suffer from visual artifacts. In contrast, combining real and synthetic data preserves the visual quality inherited from real data while effectively learning interactive behaviors from synthetic data, with only minor artifacts that do not significantly affect interaction experience.

## 6 CONCLUSION

In this work, we introduced InterAvatar, a novel framework for real-time interactive portrait animation designed to overcome two key limitations of existing methods: limited interactivity due to slow inference and the lack of behavioral interaction modeling. To this end, we adapted a video diffusion transformer for real-time portrait animation, incorporating an effective appearance decoupling strategy to improve appearance consistency over long durations. We further introduced behavioral interaction prompts into portrait animation for the first time, devising new encoding and injection methods to enhance user–avatar interactivity. To support training, we developed a hybrid data annotation pipeline that systematically collects, labels, and filters real and synthetic videos with behavioral interaction prompts. Extensive experiments on standard benchmarks demonstrate that InterAvatar achieves real-time animation speeds without compromising visual quality, while qualitative results highlight its ability to generate realistic and responsive user–avatar interactions.

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

## A  PRELIMINARY

Rectified flow (Liu et al., 2022; Lipman et al., 2022) is a deterministic generative modeling approach that learns a continuous ordinary differential equation (ODE) mapping from a simple noise prior $\pi_0$ (e.g., standard Gaussian) directly to a target data distribution $\pi_1$. Formally, rectified flow constructs a velocity field $v(x, t)$ defining the ODE

$$\frac{dZ_t}{dt} = v(Z_t, t), \quad Z_0 \sim \pi_0, \quad Z_1 \sim \pi_1,$$ (2)

transporting noise samples $Z_0$ at $t = 0$ into data samples $Z_1$ at $t = 1$.

The core idea is to align the ODE trajectories with a predefined straight-line interpolation $X_t = (1 - t)X_0 + tX_1$, where $X_0 \sim \pi_0$ and $X_1 \sim \pi_1$. Although this interpolation itself is non-causal (as it requires knowing $X_1$ in advance), rectified flow rectifies this by training the velocity field $v(x, t)$ to approximate the interpolation's velocity $(X_1 - X_0)$ without future knowledge:

$$\min_v \ \mathbb{E}_{X_0 \sim \pi_0, X_1 \sim \pi_1} \left[ \int_0^1 \left\| v\big((1 - t)X_0 + tX_1, \, t\big) - (X_1 - X_0) \right\|^2 dt \right].$$ (3)

This flow matching objective ensures that the distribution of ODE solutions $Z_t$ matches the interpolation distribution at every $t$, effectively yielding the optimal vector field

$$v(x, t) \approx \mathbb{E}[X_1 - X_0 \mid X_t = x].$$ (4)

Rectified flow provides several advantages over conventional diffusion-based generative models: it is inherently deterministic, avoids stochastic forward diffusion, and enables efficient sampling by solving a simple ODE initial value problem. Additionally, the training procedure is simulation-free, stable, and straightforward, directly optimizing the velocity field via regression without complex reparameterizations or likelihood computations.

## B  DATA CURATION DETAILS

### B.1  ANNOTATION FOR SYNTHETIC VIDEOS

**Body Region.** We define a comprehensive list of body regions for interaction, including: abdomen, elbows (left/right), calves (left/right), chest, ears (left/right), eyes (left/right), face (left/right), hair, mouth, neck, nose, hands (left/right), shoulders (left/right), and thighs (left/right). For the hair region, we first employ Qwen2.5-VL-7B to localize the area, followed by the application of SAM2 (Ravi et al., 2024) for precise hair segmentation. The center point of the segmented region is selected as the position for the point prompt. For other body regions, we use the DWPose pose estimator to obtain whole-body keypoints, and map the selected body region to its corresponding

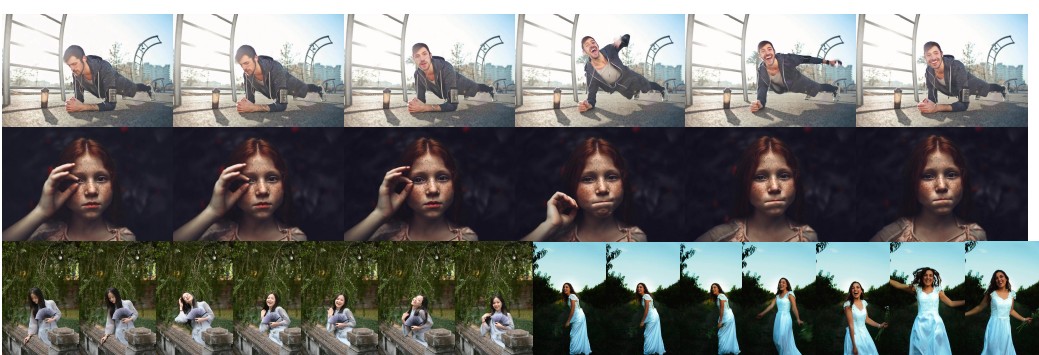

Figure 8: Ramdomly selected training samples.

coordinate based on the pose estimation results. During the annotation process, only visible body regions are considered. For instance, if the hand is occluded or not visible in the image, it is excluded from the list of annotated regions.

**Interaction Type.** We categorize interaction types into four groups: Affectionate (e.g., kissed and caressed), Gentle (e.g., stroked and blown on), Neutral (e.g., pressed and tapped), and Aggressive (e.g., punched, poked, kicked, pinched, and scratched).

**Dataset Visualizations.** We visualize some synthetic samples in Figure 8.

**GPT Prompt.** We present the GPT5-mini prompt used for synthetic data generation in Table 5.

### B.2 FILTERING FOR TALKING-HEAD VIDEOS

During data pre-processing, we apply a three-stage filtering pipeline: single-speaker extraction, motion filtering, and post-processing. First, we select videos with a single speaker to reduce background noise and scene variation, using existing tools (Plaquet & Bredin, 2023). We then apply filtering techniques to ensure the quality of head motion, head pose, and camera stability (Karaev et al., 2024; Chung & Zisserman, 2016), computing metric scores for each clip to support flexible screening strategies. Finally, we extract a random frame from each video and encode facial embeddings with InsightFace (Ren et al., 2023), while audio tracks are encoded with Wav2Vec2 (Baevski et al., 2020) to provide multimodal conditions during training.

## C IMPLEMENTATION DETAILS

### C.1 TRAINING.

**Wan2.1-1.3B Baseline.** We train the Wan2.1-1.3B baseline on 64 Nvidia H800 GPUs with a batch size of 64 and an initial learning rate of 2e-5 under a cosine decay schedule. Training proceeds in three stages: (1) 80K steps on audio-driven videos, (2) 40K steps on a mixture of audio-driven and behavioral interaction videos, and (3) a distillation stage that follows the Self-Forcing (Huang et al., 2025) setting. A multi-scale training strategy is employed, resizing each frame to 480p with varying aspect ratios. The number of frames per video ranges from 25 to 101. During training, conditioning inputs are randomly dropped with probability 0.1, while all conditioning signals are simultaneously dropped with probability 0.05. For behavioral interaction videos, captions are dropped with probability 0.5. We use InterAvatar with Wan2.1-1.3B as the default model for ablation studies.

**LTX-Video Baseline.** The LTX-Video baseline is trained under the same setup but with a batch size of 128. The two training stages consist of (1) 120K steps on audio-driven videos, and (2) 40K steps on the mixed dataset. Since the original LTX-Video is designed for real-time generation, the Self-Forcing stage is not applied. The frame resolution is consistently resized to 480p with variable aspect ratios, and video lengths range from 37 to 129 frames. The same conditioning dropout scheme is applied: individual inputs are dropped at probability 0.1, all signals jointly at 0.05, and captions for behavioral interaction videos at 0.5.

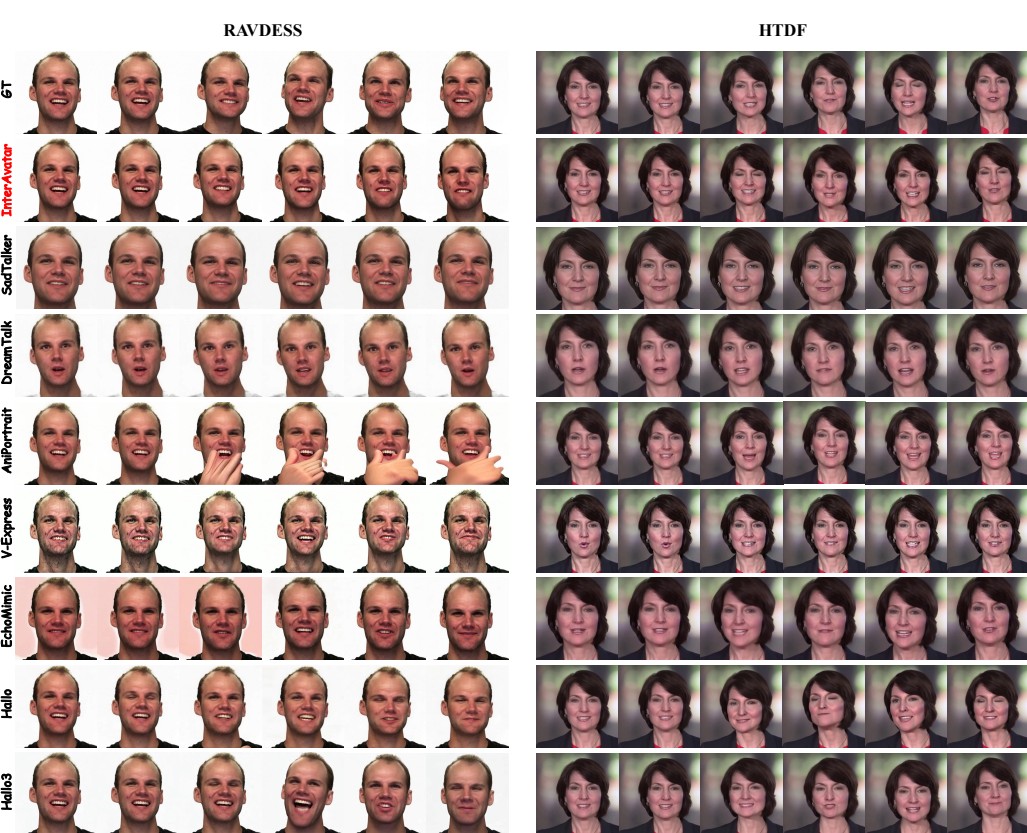

Figure 9: Qualitative comparisons on the RAVDESS and HDTF dataset.

Table 5: **Guidelines for the interaction simulation prompt generation.** Each block summarizes one aspect of the prompt that governs how the model should generate context-aware reactions.

---

**Scenario Overview**

You will analyze an image of a person and responding to hypothetical interactions by fully embodying their physical and emotional presence. Each hypothetical interaction specifies a target body part and an interaction type. The person is always facing the camera while reacting, and all interactions occur as if the user is positioned directly in front of them, just outside the camera's view. You will act as if you are this person, responding purely based on their internal and physical reactions to the situation, while drawing from the environmental context (e.g., clothing, setting, items held). Your action and emotion should be reasonably expressive instead of cold and static.

---

**Independent Reactions**

(1) The Action must describe how the person physically reacts, including instinctive movements and physical adjustments. Avoid describing the external source of contact.
(2) Responses must exclude any mention of external initiators or interactions and assume the person is facing forward toward the camera at all times.
(3) When eyes, ears, limbs, or other paired features react, explicitly state *left* or *right*.

---

**Narration Style**

All descriptions must consistently use "The person/man/woman/child/boy/girl" instead of pronouns like "they", "their", or "I".

---

**Emotion Intensity Scale**

Use the following terms to indicate emotional scale:
(1) mild: Subtle reaction, slight emotion.
(2) moderate: Clear reaction, visible emotion.
(3) strong: Pronounced reaction, intense emotion.

---

**Environmental Contextualization**

Analyze the person's clothing, held items, and setting from the image to ensure contextual accuracy. The person's response must reflect their physical state, posture, and surroundings. Assume they are holding their position and facing the camera at all times. Consider the environmental context (e.g., items held) to provide a more natural and contextually appropriate response.

---

**Body Part Inclusion**

(1) Provide reactions for every listed body part that appears in the frame.
(2) Exclude any part that is not visible, without altering the output structure.

---

**Expressive Detail**

(1) Paint vivid micro-expressions—e.g., eyebrow lifts, lip twitches, or soft exhales.
(2) Pair physical movement with a matching emotional tone to convey authenticity.

---

## C.2 EVALUATION.

When comparing our method with other methods, we apply spatiotemporal skip guidance (Hyung et al., 2024). The reference image is used as the initial motion frame, while the last 5 frames of each generated clip are used to condition the subsequent clip. Evaluations are performed on HDTF (Zhang et al., 2021), CelebV-HQ (Zhu et al., 2022), and RAVDESS (Livingstone & Russo, 2019), using 100 randomly sampled videos from each dataset. We evaluate visual fidelity using Fréchet Inception Distance (FID) (Heusel et al., 2017) and Fréchet Video Distance (FVD) (Unterthiner et al., 2019). FID measures the similarity between the distribution of generated and real images and FVD extends this to the temporal domain by evaluating coherence and realism across video sequences. Following FantasyTalking (Wang et al., 2025b), we measure identity consistency (IDC) and aesthetic quality (ASE). Additionally, we use SyncNet (Chung & Zisserman, 2017) to compute the Sync-C score, which quantify the lip-audio synchronization between lip movements of the generated avatar and the input audio.

Figure 10: Comparison of different interaction positions. The interaction type is kick. The red circle marks the interaction region in the first column, while the blue circle indicates the interaction region in the second column.

## D    MORE RESULTS

### D.1    QUALITATIVE COMPARISONS WITH EXISTING METHODS

We conduct qualitative comparisons between InterAvatar and existing methods. As shown in the Figure 9, InterAvatar achieves promising results with the strongest appearance consistency. In contrast, rendering-based methods such as DreamTalk and SadTalker often exhibit unnatural head jittering, although this may be less noticeable when visualized as individual frames.

### D.2    DISCUSSION

**Prompting Strategy Comparison.** Text prompts exploit the pretrained video–text alignment of the diffusion model, which improves semantic comprehension. However, this approach is impractical for interactive settings, as users must carefully craft detailed descriptions and manually input them, a process that is time-consuming and lacks interactivity. In contrast, our proposed behavioral interaction prompts require additional architectural design and training to unlock this capability, but they deliver a much more intuitive user experience. With a simple click on an interested region and a keyword defining the interaction type, the avatar reacts immediately, enabling real-time responsiveness and a more natural interaction paradigm.

### D.3    ABLATION STUDY

**Positions of Interaction Prompts.** We investigate how the spatial location of interaction prompts affects the generated avatar responses. Since reactions are intrinsically tied to the interacted region, changing the prompt position leads to distinct avatar behaviors, as illustrated in Figure 10. Notably, when keeping the interaction type and reference image fixed, the model still produces contextually appropriate but different reactions depending on the chosen position.

## E    THE USE OF LARGE LANGUAGE MODELS

We used large language models (LLMs) in limited ways that did not contribute to the scientific content or conclusions of the paper. Specifically, an LLM was employed for writing polish and editing, helping to improve grammar, clarity, and readability of the manuscript. In addition, LLM-based large multimodal models (LMMs) were used for data curation and data filtering.

## F    LIMITATIONS

We acknowledge some limitations in our paper that require attention. One limitation is that we only regard the human body region as valid for interaction, and clicks outside of this region, such as in the background, are not properly handled. This may lead to unexpected behavior or degraded animation quality due to ambiguous input signals that fall outside the model's defined interaction scope. In future work, we plan to extend our interaction framework to include contextual interaction about the entire scene, enabling more robust responses to diverse user inputs.

