# OpenReview forum: "InterAvatar: Real-time Interactive Portrait Animation via Behavioral Interaction Prompts"
_ICLR.cc/2026/Conference — ICLR 2026 Conference Withdrawn Submission_

### Official Review · Reviewer_eWRu · 2025-10-25

**Soundness:** 2
**Presentation:** 2
**Contribution:** 1
**Rating:** 2
**Confidence:** 4

**Summary:**

This paper proposes InterAvatar, a real-time interactive portrait animation framework designed to address two critical limitations of existing methods: slow inference speeds (incompatible with interactive scenarios) and the absence of behavioral interaction capabilities. Built on lightweight video diffusion transformers (Wan2.1-1.3B and LTX-Video-2B) optimized via diffusion distillation, InterAvatar conditions animation on three inputs: a reference image, audio signals, and behavioral interaction prompts (user-specified body regions + interaction types, e.g., "touch left shoulder"). Evaluated on HDTF, CelebV-HQ, and RAVDESS, InterAvatar achieves state-of-the-art performance (e.g., FID=11.171 vs. 15.820 for Hallo3; Sync-C=6.072 vs. 1.935 for Hallo3) while maintaining real-time speed: 80 frames (512×512) generated in 5 seconds on an NVIDIA H800 GPU. Qualitatively, it simulates natural responses to behavioral prompts (e.g., a "punch" to the nose triggers a pain reaction).

**Strengths:**

- Most DiT-based video diffusion models (e.g., Wan2.1-14B, Wang et al. 2025a) require ~30 minutes for a 5-second clip, making them unusable for interactive applications. InterAvatar’s optimization (distillation + lightweight backbones) achieves 16 FPS (80 frames/5s) on an H800—30× faster than Hallo3 (Cui et al. 2024) at the same resolution—without sacrificing quality (FVD=87.680 vs. 115.060 for Hallo3).
- Existing methods (e.g., AniPortrait, Wei et al. 2024; SadTalker, Zhang et al. 2023b) only support audio/text control, lacking physical interaction. InterAvatar is the first to introduce behavioral interaction prompts, enabling intuitive user-avatar interactions (e.g., click + interaction type). This is validated by qualitative results (Fig. 5) showing context-aware reactions (e.g., "pinch left eye" triggers hand-to-eye movements) and quantitative gains in interaction realism (not measured by baselines).
- Methods like SadTalker and DreamTalk (Zhang et al. 2023a) mix identity and attribute features, leading to temporal drifting. InterAvatar’s decoupled representation (identity via InsightFace, attributes via DINOv2) outperforms single-representation approaches: IDC=0.981 (vs. 0.972 for identity-only, 0.967 for attribute-only) and FVD=87.680 (vs. 121.114 for identity-only), ensuring long-video consistency.
- Public datasets (e.g., HDTF, CelebV-HQ) lack user-avatar interaction data. InterAvatar’s hybrid pipeline (real + synthetic data) addresses this: real videos (OpenHumanVid) ensure visual fidelity, while synthetic videos (GPT-generated prompts + Wan2.2) teach interaction patterns. Ablations show this outperforms real-only (artifacts) or synthetic-only (low fidelity) training (Fig. 7).

**Weaknesses:**

- Training requires 64 NVIDIA H800 GPUs (Wan2.1-1.3B) or 128 H800s (LTX-Video), which is inaccessible to most research teams. Inference also depends on high-end GPUs (H800)—no evaluation on edge devices (e.g., RTX 4090) limits real-world deployment. This contrasts with lightweight methods like StrucBooth (trained on a single 48G A6000 GPU).
- Only body regions (e.g., eyes, shoulders) are valid for interaction; clicks on backgrounds trigger no meaningful response (acknowledged as a limitation). This lags behind methods like Follow-Your-Click (Ma et al. 2024), which support open-domain regional animation.
- Only 4 coarse categories (Affectionate, Gentle, Neutral, Aggressive) are supported—no fine-grained interactions (e.g., "tap softly" vs. "tap firmly") or dynamic inputs (e.g., mouse drags, multi-region clicks).
- Despite focusing on "interactive experience," the paper only reports objective metrics (FID, Sync-C) and qualitative examples. SOTA interactive works (e.g., Durian) include user studies to validate perceived realism and usability—this absence weakens claims of "enhanced interactive user experience."
- The synthetic data pipeline relies on GPT to generate interaction prompts, but the paper provides no details on validating prompt consistency (e.g., whether "punch" always elicits pain reactions) or filtering unrealistic synthetic outputs. This raises concerns about training data quality and generalization.
- InterAvatar ignores interactions with background elements or objects held by the avatar (e.g., clicking a cup the avatar holds). This limits immersion—real-world interactions often involve both human bodies and their surroundings. Methods like Playable Environments (Menapace et al. 2022) already support scene-level interaction, highlighting this gap.
- The paper does not test if reactions are consistent with avatar context (e.g., a child vs. an adult reacting to "pinch"). A child might cry, while an adult might flinch—but InterAvatar’s current design may generate generic responses, reducing realism.
- Training data focuses on real humans—no evaluation on stylized avatars (e.g., cartoons, 3D characters) or non-human subjects. This contrasts with methods like Stable Video-Driven Portraits (from prior reviews), which generalize to sketches and Ghibli-style inputs.

**Questions:**

- Can InterAvatar be optimized for edge GPUs (e.g., NVIDIA RTX 4090, Jetson AGX) to enable deployment on consumer hardware? What is the trade-off between speed and quality on low-end devices?
- How would the model distinguish between semantically similar interactions (e.g., "tap" vs. "poke")? Would adding fine-grained interaction labels (e.g., force intensity) improve response specificity?
- How were GPT-generated prompts validated to ensure they produce contextually appropriate reactions? Were human annotators used to filter unrealistic prompts or synthetic videos?
- Would pre-training on stylized/non-human data enable InterAvatar to animate cartoon or 3D avatars? Does the appearance decoupling strategy transfer to non-human identity/attribute features?

---

### Official Review · Reviewer_kfz6 · 2025-10-31

**Soundness:** 2
**Presentation:** 3
**Contribution:** 2
**Rating:** 4
**Confidence:** 5

**Summary:**

InterAvatar introduces a real-time interactive portrait animation framework using behavioral interaction prompts. It adapts video diffusion transformers (Wan2.1-1.3B, LTX-Video-2B) for multi-condition generation, incorporating a representation decoupling strategy to enhance appearance consistency. A hybrid data curation pipeline supports training with annotated real and synthetic videos. Evaluations show comparable video quality to state-of-the-art models and effective simulation of behavioral interactions, achieving 80 frames in 5 seconds on an H800 GPU.

**Strengths:**

The strengths of this paper lie in the following aspects:
1) Modeling the user interaction in the portrait animation is instresting. First integration of behavioral interaction prompts into portrait animation. Hybrid encoding combines (x, y) positional tokens (MLP-projected) with CLIP visual prompting via a red marker, and text injection via umT5 cross-attention using a templated sentence—yielding precise, context-aware reactions (Sec. 3.4).
2) This work shows promising real-time DiT adaptation. Practical pipeline on Wan2.1-1.3B/LTX-Video-2B with diffusion distillation, 3D RoPE, conditioning dropout, and audio cross-attention (wav2vec); 80 frames at 512×512 in ~5 s on H800. Previous approaches animation approaches, MimiMotion and Hallo series failed to.
3) The work also propose the data curation at scale, for this interesting user interaction based portrait animation approach. Hybrid real+synthetic interaction videos with automated labeling (Qwen2.5-VL + DWPose/SAM2), action/emotion prompt generation (GPT), and strict filtering (CoTracker motion, CLIP/InsightFace temporal/identity consistency, anatomy checks) to teach avatar–user interaction patterns (Sec. 4).
4) Finally, the presentation of this paper is straightforward, very easy to follow and understand. I donot doubt the reproductivity of this approach.

**Weaknesses:**

My major concern lie in the fact that:
this work seems to be an engeering combination of exsiting skill. The talking head framework seem to base on the DiT framework, adopted by Hallo3, except that hte FaceNet together with DINOv2 is adopted the keep the face identity. The audio are embedded with Wav2Vec and injected as a condition to DiT diffuser through cross attention. The only difference is the user input, which adopts CLIP and umT5 for embedding. This work demonstrates more innovation in face animation at the interaction level than in methodology, so it is not suitable for a conference like ICLR.

Some other technical questions:
1) Interaction evaluation is qualitative; no quantitative metric or user study for behavioral response realism/alignment.
2) Interaction scope is narrow. Limited to single visible person, single-click body-region prompts; clicks outside body are unsupported; multi-step/multi-click and continuous interactions are not studied.
3) Baseline coverage is limited to audio-driven/talking-head methods; lacks quantitative comparisons to interactive/local-control video methods (e.g., Follow-Your-Click) and broader DiT baselines under identical settings.
4) Ethics/data governance are recommended to discuss, at least a short paragraph. Use of web-scale human videos and synthetic aggressive interactions without detailed discussion of consent, licensing, demographic bias, or misuse risks.

**Questions:**

Please see the weakness section.

---

### Official Review · Reviewer_GWeA · 2025-10-31

**Soundness:** 3
**Presentation:** 3
**Contribution:** 2
**Rating:** 4
**Confidence:** 3

**Summary:**

The paper proposes InterAvatar to generate real-time interactive portrait animation which supports behavioral interaction prompts. The method aims to address the slow inference speed and lack of interaction modeling limitations in existing video diffusion-based methods. Specifically, they leverage the diffusion distillation to support real-time animation, in which introduces the decoupling between identity and attribute representation to enhance appearance stability. And it uses behavioral interaction prompts consisting of spatial click coordinate and textual action descriptions to trigger context-aware avatar reactions. For the training data, the paper presents data curation pipeline to collect and filter real and synthetic video data.

**Strengths:**

This paper is the first work to introduce behavioral interaction prompts into video diffusion-based portrait animation system. The idea of involving user interaction into facial animation is overall novel and interesting. To achieve this goal, the authors use the combination of real and synthetic dataset pipeline to ensure both realism and interaction variety. Also, the model achieves real-time generation using a DiT architecture, avoiding slow inference limitation in previous methods. Experiments in paper show improved identity consistency, temporal stability, and user-avatar interaction expressiveness compared with SOTA methods. Moreover, detailed network architecture, data processing and training mechanisms are presented, thus the paper is overall clear and easy to follow.

**Weaknesses:**

1. The animation videos in supplementary materials have artifacts, especially on the inner mouth area.
2. No video comparisons with Hallo3 shown in the submission, making it hard to evaluate the lip motion accuracy and appearance stability. Moreover, there are no user study or ablation video results, which which would strengthen the claim and evaluation.
3. Figure 7 indicates that training only on real data lacks interactivity, which means the interactive behaviors largely relies on synthetic data generated by Wan2.2. This raises concerns on whether the learned behavior patterns are natural and accurate.
4. The expressiveness of the interaction behavior are constrained by the pre-defined body region and interaction types in the synthetic dataset.

**Questions:**

Only audio-driven results or interactive videos are shown in the supplementary video. Can the model support behavioral interaction while being driven by the audio? If so, I recommend the authors showing such video results. In addition, the artifacts in the video appears to come from the diffusion distillation for real-time animation. It would be helpful if the authors show more qualitative results on the decoupling of identity and attribute for appearance enhancement.

---

### Official Review · Reviewer_4e4C · 2025-10-31

**Soundness:** 2
**Presentation:** 2
**Contribution:** 2
**Rating:** 4
**Confidence:** 4

**Summary:**

This paper presents a novel framework, InterAvatar, for real-time and interactive portrait animation. The core contributions are: (1) A "behavioral interaction prompt", which conditions the generation on both a spatial location (e.g., a click) and an interaction type (e.g., 'touch' or 'punch'). (2) A representation decoupling strategy that separates identity (using FaceNet) and attribute (using DINOv2) embeddings to improve appearance consistency and reduce temporal drifting in real-time models. (3) A hybrid data curation pipeline to source, generate, and annotate training data for this new interaction task. The authors demonstrate that InterAvatar achieves competitive performance on standard video quality benchmarks (e.g., FID, FVD) while operating at real-time speeds (80 frames in 5 seconds on an H800).

**Strengths:**

1. The paper proposes a novel  task: simulating behavioral reactions rather than just user-defined motion. The authors systematically address the key challenges, from the novel conditioning (the prompt) to the data requirements (the pipeline) and the model architecture (the decoupling strategy).

2. The qualitative experiments (e.g., Figure 1 and Figure 5) effectively demonstrate the potential of the behavioral interaction prompt. The model shows a clear ability to generate contextually different and appropriate reactions (e.g., a 'punch' to the 'nose' generates a pain/crying response) , which is a compelling proof-of-concept for this new interaction paradigm.

3. The paper includes a strong set of ablation studies that validate key design choices. Specifically, the ablations for appearance encoding (Table 3) , injection methods (Table 4) , and the hybrid data strategy (Figure 7)  clearly support the final architectural decisions.

**Weaknesses:**

1.The paper's primary claim is the "behavioral interaction". However, this core contribution is evaluated only qualitatively. There is no quantitative benchmark, metric, or user study provided to measure the quality of the interaction. Key questions remain unanswered: How "natural" or "appropriate" are the reactions? How diverse are the responses to the same prompt? Without this, the effectiveness of the main idea is not rigorously validated.

2.While the paper proposes a representation decoupling strategy to reduce drift, the evaluation of this specific component feels incomplete. The ablation in Table 3 shows the components are useful, but it does not benchmark the final strategy against other strong appearance-preservation techniques from related work. A dedicated experiment measuring identity drift over longer sequences (e.g., comparing identity similarity at frame N vs. frame 1 is needed to substantiate the claims of "enhancing appearance consistency", the authors may refer some previous work, e.g., OmniHuman.

**Questions:**

1. What is the advantage of behavioral interaction prompt compared with other modeling methods like mouse click on the realtime interative video generation model?  Would it be better to use others interaction like Follow-Your-Click (Ma et al., 2024)?

2. How does the model performs compared with other realtime video generation models in terms of video quality, id-perserving?


If the authors can solve my concerns, I will consider to raise my score.

---

### Note · Authors · 2025-11-12

I have read and agree with the venue's withdrawal policy on behalf of myself and my co-authors.